# Evaluation of the Effects of Enlarged Housing on Social Play and Reward Seeking in Rats

**DOI:** 10.3390/ani15182757

**Published:** 2025-09-22

**Authors:** E. J. Marijke Achterberg, Anne-Marie J. M. Baars, Daphne A. van Hal, Heidi M. B. Lesscher, Pascalle L. P. Van Loo

**Affiliations:** 1Department of Population Health Sciences, Unit of Animals in Science and Society, Faculty of Veterinary Medicine, Utrecht University, Yalelaan 2, 3584 CM Utrecht, The Netherlands; e.j.m.achterberg@uu.nl (E.J.M.A.); j.m.baars@uu.nl (A.-M.J.M.B.); daphne.vanhal@hvhl.nl (D.A.v.H.); h.m.b.lesscher@uu.nl (H.M.B.L.); 2Animal Welfare Body Utrecht, Utrecht University, Bolognalaan 50, 3584 CJ Utrecht, The Netherlands

**Keywords:** rats, animal welfare, housing, behavioural needs, social play, reward-sensitivity

## Abstract

This study examines the effects of a larger, enriched cage system (EC4Rats) on social play and reward-seeking behaviour in rats in research settings. Traditional cages limit natural behaviours like running and climbing, but the EC4Rats design allows for more natural activity. The study found that while there were slight changes in social play, reward sensitivity and motivation for sucrose were unaffected. Overall, the findings support the use of more spacious and naturalistic housing without compromising behavioural research outcomes.

## 1. Introduction

The ability for captive animals to perform natural behaviour has been widely recognized as a prerequisite for good animal welfare [1,2]. The European directive 2010/63/EU [3] states that ‘all animals should be allowed adequate space to express a wide behavioural repertoire. Animals should be socially housed where possible and be provided with an adequately complex environment within the animal enclosure to enable them to carry out a range of normal behaviours.’

The established legal minimum cage dimensions for medium-sized lab rats (Wistar male +/− 8 weeks old: 300 g, body length without tail is about 21–23 cm, height while walking is about 8 cm) are 800 cm^2^ ground floor area and 18 cm height (2010/63EU, Appendix III [3]), a size roughly comparable to a shoebox (smallest commercially available rat cage: l × w × h, 42.5 × 26.6 × 18.5 cm Makrolon^®^ Eurostandard type III). This legal minimum is restricting, preventing rats to perform many of their natural behaviours. For example, rats are unable to run, stand upright, dig, or climb, and nest-building is limited in standard laboratory rat housing systems [4,5]. For rats, it has been shown that the drive to perform these behaviours is innate, despite decades of breeding in captivity [6], and a range of studies investigating the preference of rats for bedding, shelter, complexity and segregation of space show the need for more space to accommodate this (see [7] for a review). In addition, several studies in rodents indicate that restricted space leads to physically less healthy animals [8,9,10]. This clearly indicates that regarding rat housing, Appendix III of the EU Directive [3] seriously falls short of its own recommendations.

Several cage suppliers now market cages in which rats have more space to perform their natural behaviours. Tecniplast, for example, markets the Double Decker and Emerat, and Innovive markets the Tall and Tall XL cages. These cages are substantially higher and wider than the standard Makrolon^®^ III or IV cages. However, the design of these cages is based on enlarging existing cages, and they have become heavy and less manageable as a result. Hence, there is a limit to increasing the size of these systems and therefore also for the desired cage measures for welfare improvement in rats. The EC4Rats, a cage system that was designed by Scanbur and is currently in development, circumvents this logistical bottleneck. This system is both more manageable and large enough for rats to express a wide range of behaviours. The difference between this system and other larger housing systems is that the rats are taken out of the cage, rather than the cage being taken out of the rack, comparable to standard rabbit housing.

To facilitate the use of larger cage systems it is important to determine the impact of such larger housing systems, in which more natural behaviour can be displayed, on behavioural outcomes. This is important because previous studies have shown that environmental enrichment, for instance, affects behavioural outcomes such as anxiety, learning, memory, social behaviour and reward seeking in both rats and mice [11,12,13,14,15,16]. In this study, we therefore sought to compare behavioural data for rats that were housed in regular cages with mandatory enrichment to those that were housed in the larger EC4Rats cages with either mandatory or additional enrichment. A prototype of the system has been tested with promising results [14]. In this project, we extended this work and determined the impact of housing rats in the EC4Rats system (Scanbur Karlslunde, Denmark) on key research output parameters in our facility, i.e., social play behaviour, reward sensitivity and motivation.

1.Social play behaviour:

Play is rewarding and important for the development of brain and behaviour of a wide range of species. During social play, young mammals experiment with their (social) behavioural repertoire, thereby facilitating the development of social competence, emotion regulation, resilience to stressful stimuli, cognitive flexibility and problem-solving skills [17,18,19,20]. Studies in rats have been imperative to our understanding of the role of social play behaviour in the development of brain and behaviour. Play restriction during the developmental period when social play is most abundant, (i.e., postnatal day 21–42 [21,22,23,24,25]), effectively depriving rats from social play behaviour, result in brain alterations but also social and cognitive impairments, evident from altered social behaviour, impaired impulse control and decision-making. The degree to which social play is expressed is highly dependent on environmental conditions. Stress but also enriched environments both reduce social play behaviour [26,27]. Therefore, in this project we sought to determine the effect of housing in the EC4Rats on the duration and structure of social play behaviour compared to conventional housing.

2.Reward sensitivity and motivation

To study complex behaviours, ranging from social choice, empathy, cognitive control and flexibility, impulse and inhibitory control, reward sensitivity to addictive behaviour, we rely heavily on operant tasks. In operant chambers, rodents (often rats or mice) can respond (i.e., press a lever or make a nose-poke) for a rewarding stimulus (sucrose, substance of abuse, or a social partner) under various schedules of reinforcement. Importantly, environmental factors such as enriched housing conditions have been shown to affect reward sensitivity and cognitive functioning and the neurobiological substrates involved in these behavioural domains [28,29]. Therefore, it is important to understand the impact of alternative housing conditions on these behaviours to safeguard the validity and continuity of ongoing and future research projects.

The aim of the study was to determine whether housing animals in larger cages with mandatory enrichment or with additional enrichment would affect play behaviour or sucrose reward sensitivity. We hypothesized that housing rats in larger cages would reduce their tendency to engage in social play but would reduce their sensitivity and motivation for sucrose rewards.

## 2. Materials and Methods

### 2.1. Housing, Animals, Husbandry and Enrichment

In this study, 48 male Lister Hooded rats (Charles River, Sulzfeld, Germany) were used that were at the age of PND 21–23 and weighted 44.3 ± 0.99 g (mean ± SEM) at the start of the experiment. The animals were housed in three different housing systems in temperature-controlled rooms (20–21 °C, 60–65% relative humidity), with ad libitum access to food and water, under a 12:12 h light–dark schedule (lights on at 19.00 h). All experiments were performed during the dark phase of the day–night cycle. The procedures performed in the study are non-invasive with the exception of individual housing of the young animals for a period of one week. The study was carried out under project licence AVD1080020197545 issued by the Central Committee for Animal Experiments in The Netherlands. A detailed study protocol was prepared in advanced and formally reviewed and approved by the Animal Welfare Body.

Upon arrival in the facility, the animals were randomly distributed over the following three experimental conditions:Standard housing (Makrolon^®^ IV, Tecniplast, Varese, Italy) with standard (mandatory) enrichment (Perspex shelter (Techniplast, Varese Italy), Bulkysoft tissues (Lyreco, Marly, France) and an Aspen gnawing stick (Bioservices, Schaijk, Netherlands), *N* = 16.EC4Rats (Scanbur, Karlslunde, Denmark) with standard (mandatory) enrichment (elevated platform, Perspex shelter (Techniplast, Varese Italy), Bulkysoft tissues (Lyreco, Marly, France) and an Aspen gnawing stick (Bioservices, Schaijk, Netherlands)), *N* = 16.EC4Rats with EC4Rats enrichment (Elevated platform, ladder, hammock, tube and hanging wooden block for gnawing, Scanbur, Karlslunde, Denmark), *N* = 16.

A prototype of the EC4Rats was made available by Scanbur for this project. The prototype consisted of a ventilated cabinet with nine rat cages, (l × b × h: 50 × 38 × 38 each, see Figure 1). The basic design is comparable to laboratory rabbit caging, with the cage entrance at the front and a separate tray for cage cleaning. The EC4Rats enrichment (an elevated platform, a hammock, a tube and a hanging wooden block for gnawing) was also made available by Scanbur.

The rats were housed in groups of four animals per cage and were allowed to acclimatize to the facility for one week. To reduce distress, animals were habituated to handlers and procedures. Subsequently, the rats were subjected to play behaviour tests (described below). Thereafter, the rats were trained for operant sucrose self-administration, as explained in more detail below. The cage compositions did not change during the experiment and food was always provided ad libitum to prevent social stress.

### 2.2. Assessment of Social Play Behaviour

Each animal was habituated twice to the testing arena (l × w × h: 40 × 40 × 60 cm, Figure 2A) for 10 min on PND 23 and 24. Thereafter, the rats were tested five times for their tendency to engage in social play behaviour, on postnatal days (PND) 27, 29, 31 (week 1) and on PND 34 and 36 (week 2), respectively. On each test day, every rat was isolated for 2.5 h prior to testing, in a Makrolon^®^ type III cage (Tecniplast, Vares, Italy) in a room different from the housing room, to increase their motivation for social play behaviour to half maximal levels and prevent floor and ceiling effects [30,31,32,33]. On test days, the rats were paired such that they could all play with an unfamiliar partner from the same experimental group of approximately the same weight (maximal difference of 10 g). For the play test, a pair of rats was placed in the testing arena for 15 min (Figure 2A). Each play session was recorded using a digital camera (Logitech ^®^, Lausanne, Switserland). The behaviour of both rats was assessed afterwards by a trained observer using Observer XT 16.0 software (Noldus Information Technology, Wageningen, The Netherlands). Testing order was stratified over groups to exclude time of day as confounding effect. The person scoring play behaviour was blinded for housing conditions.

In rats, a bout of social play behaviour starts with one rat soliciting another animal to play by pouncing, i.e., touching the nape of the neck of the other animal with its snout. If the animal that is pounced upon fully rotates to its dorsal surface with the other animal nuzzling or grooming the ventrum, ‘pinning’ is the result. From this position, the supine animal can initiate another play bout, by trying to gain access to the other animal’s neck. Thus, during social play, pouncing is considered an index of play solicitation, while pinning functions to prolong the play bout [34,35,36]. Pinning and pouncing frequencies are considered the most characteristic parameters of social play behaviour in rats [34,37]. The following behaviours were scored:

Frequency of pinning: One animal lying with its dorsal surface on the floor with the other animal standing over it, nuzzling or grooming the ventrum.

Frequency of pouncing: One animal attempting to nose or rub the nape of the other animal.

Duration of social exploration: One animal sniffing or grooming any part of the partner’s body.

Duration of non-social exploration: Moving around (walking or rearing) or sniffing any part of the test cage.

After this 2-week testing period, 24 rats were offered for re-use or adoption. The remaining 24 rats were housed in pairs with one of their previous cage mates and remained in the same experimental condition.

### 2.3. Behavioural Assessment of Operant Conditioning for Sucrose

Subsequently, when the rats were 7 weeks of age, they were trained and tested in operant conditioning chambers (29.5 × 24 × 25 cm; Med Associates Inc., Fairfax, VT, USA, Figure 2B) equipped with two retractable levers (4.8 × 1.9 cm; ENV-112CM) and a white cue light (28 V, 100 mA; ENV-221 M) present above each lever. A recessed food receptacle was situated between the levers, equipped with an infrared beam for nose-poke detection. The wall on the opposite side of the box contained a white house light (28 V, 100 mA; ENV-215 M). The floor of the chamber was covered with a metal grid with bars separated by 1.57 cm. All chambers were situated in light- and sound-attenuating cubicles equipped with a ventilation fan and were controlled by MED-PC IV software (version 4.2) for Windows.

All rats were trained to press a lever for sucrose in 30 min operant sessions, once daily, 4–5 days per week. The house light was illuminated throughout the session. The positions of the active (leading to the reward) and inactive (no reward, no consequence) levers were counterbalanced between all rats. The animals were first trained to respond for sucrose under a fixed ratio (FR) 1 schedule of reinforcement. This means that pressing the active lever once activates a pellet dispenser that delivers a 45 mg sucrose pellet (TestDiet, USA) into the food receptacle. Simultaneous with reward delivery, both levers were retracted, and the cue light above the active lever was illuminated until 1 s after the animal entered the food receptacle. Next, the cue light was turned off, and the levers were reintroduced, signalling the start of a new trial. All inactive lever presses were recorded but were without programmed consequences. When the animals had acquired stable responding (i.e., when there was less than 25% variation in the number of rewards obtained over three subsequent sessions), the rats were trained to respond for sucrose under a progressive ratio (PR) schedule of reinforcement. In the PR sessions, the response requirement for a sucrose pellet progressively increased after each obtained reward (from 1 to 2, 4, 6, 9, 12, 15, 20, 25, etc., based on [38,39]). Each PR session ended when no subsequent reward was earned for 30 consecutive minutes. Responding under the PR schedules was deemed stable when there was less than 25% variation in the number of rewards they earned over three subsequent sessions. The breakpoint under the PR schedule of reinforcement was defined as the maximum number of presses performed in the last, successfully completed ratio. Testing order was stratified over groups to exclude time of day as confounding effect. PR outcome was automated.

### 2.4. Statistical Analysis

An a priori power analysis was performed in G-Power version 3.1 based on pedal presses during PR (least sensitive primary parameter). No animals were excluded from analysis.

Play behaviour data was analyzed using 2-way ANOVAs with housing (standard + standard enrichment; EC4Rats + standard enrichment; EC4Rats + EC4Rats enrichment) and test day as between-subject factors. Data were assumed to be normally distributed. In case of a significant interaction effect, this was followed by one-way ANOVAs for the separate days. When appropriate, subsequently post hoc unpaired Student’s T-tests with Bonferroni correction were performed.

For the analyses of operant behaviour, the number of active and inactive lever presses and breakpoints were averaged over the three sessions during which the rat reached stable responding, as described. Data were assumed to be normally distributed. The data were analyzed using one-way ANOVAs with housing (standard + standard enrichment; EC4Rats + standard enrichment; EC4Rats + EC4Rats enrichment) as the between-subject factor. Kolmogorov–Smirnov tests confirmed that each parameter was normally distributed. When appropriate, post hoc analyses were conducted using unpaired Student’s *t*-tests.

The threshold for statistical significance was set at *p* < 0.05. All data are presented as mean ± SEM. Statistical analyses were conducted using SPSS 29.0 for Windows (IBM Corp., Armonk, NY, USA).

## 3. Results

### 3.1. Play Peak Week 1

#### 3.1.1. Play Frequency and Latency

Housing condition did not alter the number of play initiations or pins in general (pouncing: F_housing_(2,63) = 0.81, *p* = 0.45; pinning: F_housing_(2,63) = 0.05, *p* = 0.95) or over days (pouncing: F_housing×day_(4,63) = 0.25, *p* = 0.91; pinning: F_housing×day_(4,63) = 0.48, *p* = 0.75). During the first week in the play peak, the number of pounces and pins changed over days regardless of housing condition (pouncing: F_day_(2,63) = 4.96, *p* = 0.01; pins: F_day_(2,63) = 3.30, *p* = 0.04). Post hoc analysis showed that compared to day 1, animals initiated less pounces on day 2 (*p* = 0.009), and that animals pinned less on day 2 compared to day 3 (*p* = 0.04). No other differences between days were found for pouncing (day 1 vs. day 3: *p* = 0.91; day 2 vs. day 3: *p* = 0.13) or pinning (day 1 vs. day 2: *p* = 0.99; day 1 vs. day 3: *p* = 0.26) (Figure 3A).

The latency to pounce or pin did not differ across housing conditions in general (pouncing: F_housing_(2,63) = 0.81, *p* = 0.45; pinning: F_housing_(2,63) = 0.05, *p* = 0.95) or across days (pouncing: F_housing×day_(4,63) = 1.21, *p* = 0.32; pinning: F_housing×day_(4,63) = 0.92, *p* = 0.46). Regardless of housing condition, rats differed in their latencies to pounce and pin across days (pouncing: F_day_(2,63) = 9.42, *p* < 0.001; pinning: F_day_(2,63) = 15.98, *p* < 0.001). Post hoc analysis indicated that the latency to pounce and pin was significantly lower on day 2 and 3 compared to the first test day (pouncing: day 1 vs. day 2: *p* = 0.01; day 1 vs. day 3: *p* < 0.001; day 2 vs. day 3: *p* = 0.68) (pinning: day 1 vs. day 2: *p* < 0.001; day 1 vs. day 3: *p* < 0.001; day 2 vs. day 3: *p* = 0.30).

#### 3.1.2. Duration of Play and Exploration

Housing condition did not alter the time spent engaged in play (F_housing_(2,63) = 0.37, *p* = 0.69; F_housing×day_(4,63) = 0.21, *p* = 0.93), social exploration (F_housing_(2,63) = 1.51, *p* = 0.23; F_housing×day_(4,63) = 0.27, *p* = 0.90), or cage exploration (non-social exploration: F_housing_(2,63) = 0.54, *p* = 0.59; F_housing×day_(4,63) = 0.76, *p* = 0.56) in general or over days. The duration of time spent on play and cage exploration differed over days regardless of housing condition (play: F_day_(2,63) = 5.78, *p* = 0.005; non-social exploration: F_day_(2,63) = 3.92, *p* = 0.03). Post hoc analysis showed that compared to day 1 and 2, rats spent more time playing on day 3 (day 1 vs. 3: *p* = 0.03; day 2 vs. day 3: *p* = 0.007), but no other differences were found (day 1 vs. day 2: *p* = 0.99). The duration of non-social exploration was longer on day 2 compared to day 3 (*p* = 0.04), whereas on the other days an equal amount of time was spent on cage exploration (day 1 vs. day 2: *p* = 0.99; day 1 vs. day 3: *p* = 0.07). The time spent on social exploration remained the same across days (F_day_(2,63) = 2.86, *p* = 0.07) (Figure 3B).

### 3.2. Play Peak Week 2

#### 3.2.1. Play Frequency and Latency

Pouncing: In the second week of the play peak, the number of pounces differed between the housing conditions in a time-dependent manner (F_housing_(2,42) = 3.21, *p* = 0.05; F_housing×day_(2,42) = 4.13, *p* = 0.02; F_day_(1,42) = 1.37, *p* = 0.25). Post hoc analysis showed that on day 5, rats housed in larger cages pounced less compared to the rats housed in conventional cages, but no other differences were found (day 5: standard vs. larger cages: *p* = 0.01; standard vs. larger enriched cages: *p*= 0.30; larger cages vs. larger enriched cages: *p*= 0.48). Being housed in the larger cages did not influence pounces made (larger cage vs. larger enriched cage: *p* = 0.48). This was not the case for day 4 (standard vs. larger cages: *p* = 0.20; standard vs. larger enriched cages: *p* = 0.99; larger cage vs. larger enriched cage: *p* = 0.29). Directly comparing the number of pounces from the same cage type between test day 4 and 5 shows that in the standard housing group, the rats pounced more on day 5 compared to day 4 (*p* = 0.03). This was not the case for the rats housed in the larger cages (*p* = 0.18) nor for the animals housed in the larger cages with the EC4Rats enrichment (*p* = 0.65) (Figure 3C).

Pins: The number of pins differed depending on housing and test day (F_housing×day_(2,42) = 4.96, *p* = 0.01; F_housing_(2,42) = 1.37, *p* = 0.27; F_day_(1,42) = 4.60, *p* = 0.04). Post hoc analysis showed that pin frequency did not differ between housing conditions on day 4 (F_housingD4_(2,21) = 0.99, *p* = 0.39), but they did on day 5 (F_housingD5_(2,21) = 4.02, *p* = 0.03). Specifically, rats that were housed in the larger cages with standard enrichment pinned less compared to rats in the conventional housing group (*p* = 0.03), whereas no other differences were found between housing conditions (control vs. larger enriched: *p* = 0.76; larger standard vs. larger enriched: *p* = 0.34). Directly comparing pin frequency from the same cage type between test day 4 and 5 shows that in standard housing, rats pinned more on day 5 compared to day 4 (*p* = 0.008). This was not the case for the larger cage (*p* = 0.13) or the larger enriched cage (*p* = 0.14) (Figure 3C).

The latency to pounce or pin did not differ between housing conditions across days. (pouncing: F_housing_(2,42) = 1.46, *p* = 0.24; F_day_(2,42) = 0.001, *p* = 0.98; F_housing×day_(2,42) = 0.89, *p* = 0.32; pinning: F_housing_(2,42) = 1.46, *p* = 0.24; F_day_(2,42) = 0.73, *p* = 0.40; F_housing×day_(2,42) = 0.56, *p* = 0.58.

#### 3.2.2. Duration of Play and Exploration

Play duration: The amount of time engaged in play differed across housing conditions over days (F_housing×day_(2,42) = 4.82, *p* = 0.01, F_housing_(2,42) = 0.04, *p* = 0.96; F_day_(2,42) = 1.50, *p* = 0.23). Post hoc analysis revealed that when housing conditions were compared per day, no differences in the time spent playing were found (F_housingD4_(2,21) = 2.70, *p* = 0.09, F_housingD5_(2,21) = 2.25, *p* = 0.13). Directly comparing play duration from the same cage type between test day 4 and 5 shows that in the standard housing group, the rats spent more time engaged in play on day 5 compared to day 4 (*p* = 0.008). There was an opposite trend for the rats housed in the larger cages (*p* = 0.07) but no difference for the rats housed in the larger cages with EC4Rats enrichment (*p* = 0.13) (Figure 3D).

Social exploration: The rats spent more time exploring each other on day 4 compared to day 5 (F_day_(2,42) = 6.76, *p* = 0.01), but there was no effect of housing condition (F_housing_(2,42) = 0.32, *p* = 0.73; F_housing×day_(2,42) = 1.44, *p* = 0.25) (Figure 3D).

Non-social exploration: The amount of time spent on non-social exploration differed per housing condition across days (F_housing×day_(2,42) = 4.48, *p* = 0.02; F_day_(2,42) = 0.009, *p* = 0.92; F_housing_(2,42) = 0.12, *p* = 0.88). Separate one-way ANOVAs per day, however, revealed that the housing conditions did not influence the amount of time spent on non-social exploration on either day 4 (F_housingD4_(2,21) = 3.28, *p* = 0.06) or day 5 (F_housingD5_(2,21) = 1.66, *p* = 0.22). Directly comparing the time spent on non-social exploration from the same cage type between test day 4 and 5 revealed that in the standard housing group, the rats tended to spend less time on non-social exploration on day 5 compared to day 4 (*p* = 0.06). The rats housed in the larger cage with standard enrichment spent more time on non-social exploration on day 5 compared to day 4 (*p* = 0.02), but no difference between these days was observed for the rats housed in the larger cages with EC4Rats enrichment (*p* = 0.63) (Figure 3D).

### 3.3. Operant Conditioning for Sucrose

Operant self-administration for sucrose under a fixed ratio 1 schedule of reinforcement did not differ between the housing conditions in the number of active lever presses (ALP, F_housing_(2,23) = 2.30, *p* = 0.13), nor in the number of inactive lever presses (ILP, F_housing_(2,23) = 0.70, *p* = 0.51) the rats made (Figure 4 and Figure 4B, respectively). In line with these findings, there were also no differences between the housing conditions in the motivation for sucrose as determined using the progressive ratio schedule of reinforcement (summarized in Figure 4C–F). There was no significant overall effect of housing on the number of active lever presses (ALP, F_housing_(2,23) = 0.10, *p* = 0.91), the number of rewards the rats earned (rewards, F_housing_(2,23) = 0.35, *p* = 0.71), nor in the breakpoint (BP, F_housing_(2,23) = 0.10, *p* = 0.90). There was, however, a non-significant trend towards an overall effect of housing on the number of inactive lever presses (ILP, F_housing_(2,23) = 3.27, *p* = 0.06) during the progressive ratio tests. Post hoc pairwise comparisons revealed that rats housed in EC4Rats cages with standard enrichment tended to make less inactive lever presses compared to rats that were housed in the EC4Rats cages with EC4Rats enrichment, but this did not quite reach statistical significance either (*p* = 0.05).

## 4. Discussion

In this study, we compared social play behaviour and operant responding for sucrose under both a fixed ratio 1 and a progressive ratio schedule of reinforcement for rats that were housed in larger housing systems to those that were housed in conventional cage systems. There were only subtle and time-dependent effects of housing condition on social play behaviour, in that the rats housed in the EC4Rats cages with standard enrichment pounced and pinned less compared to the rats housed in standard cages and compared to those housed in the EC4Rats cages with matching enrichment. This effect was only apparent for the final play test on post-natal day 36. Housing rats in larger cages did not affect the reinforcing effects of sucrose nor did it impact their motivation to seek sucrose. These findings suggest that housing systems that provide rats with more opportunities to engage in natural behaviours are compatible with reward-related research, specifically play behaviour and operant responding for sucrose.

The current findings suggest that social play behaviour, reward sensitivity and motivation for sucrose are sufficiently robust to the extent that they are not affected by housing rats in larger cages, with mandatory or with additional cage enrichment. This finding has important implications as it suggests that it is possible to house rats in larger cages without affecting important readouts. Although there is little empirical evidence to compare these findings to previous data on larger housing systems for rats, there is a wealth of literature on the impact of environmental enrichment on reward-related behaviours. For example, enrichment and housing rats in larger cages augmented social play in rats [40], particularly in subjects that were exposed to prenatal stress [41]. In line with our findings, Brenes et al. [42] described no to modest reductions in sucrose preference, depending on the age of the rats. By contrast, other studies [12,43] report reductions in operant responding for sucrose and in cue-induced reinstatement of sucrose-seeking in rats upon both acute and chronic environmental enrichment enrichments. The age of onset of environmental enrichment, i.e., in adulthood as opposed to immediately after weaning, may explain these apparent discrepancies. Future studies are required to determine the potential impact of housing rats in larger cages on more complex behaviours such as emotional contagion, social choice tasks, compulsive substance seeking, cognitive control or impulsivity.

During the study, we made notes on how the EC4Rats was used by the rats, and how the animal caretakers managed their tasks. These anecdotal observations were not formally collected in a scientific manner; however, they are interesting to mention nonetheless (personal communication with the animal care staff). We noticed that the rats used the entire cage, except for the small animals that could not reach the elevated platform (therefore, we inserted a ladder and lowered the hammock). When the rats were very small, an insert had to be added to enable the rats to reach the drinking nipple. Furthermore, the light intensity between the nine cages differed significantly. The middle cages were darker due to the closed back and bar handles at the front. We recommend the cages to be adjusted accordingly to circumvent potential impacts of such differences in light intensity on the rats’ behaviour.

## 5. Conclusions

Based on the current findings, we conclude that rats can be housed in the large EC4Rats cages without scientifically jeopardizing the animal model for reward-sensitive research (social play and food–reward). Since the paradigms used in this study are based on positive stimulation, it is possible that housing rats in larger cages might affect other types of behavioural paradigms, such as paradigms based on aversive stimulation.

## Figures and Tables

**Figure 1 animals-15-02757-f001:**
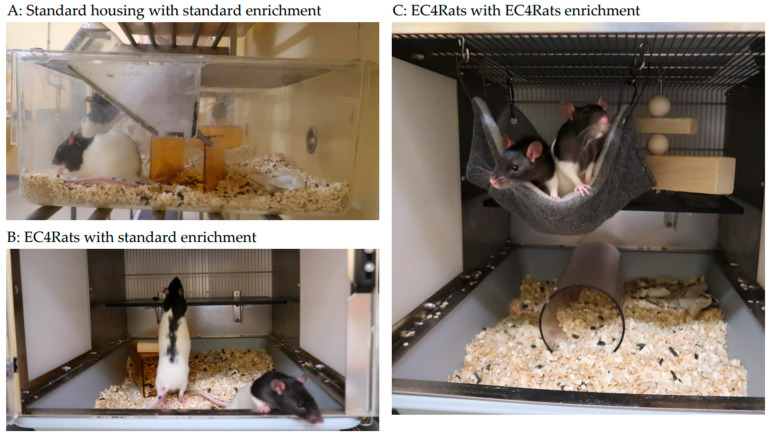
Experimental housing conditions. (**A**): standard cages with standard enrichment (Perspex shelter, tissues and a gnawing stick), (**B**): EC4Rats cages with standard enrichment and (**C**): EC4Rats cages with EC4Rats enrichment (elevated platform, ladder, hammock, tube and hanging wooden block for gnawing).

**Figure 2 animals-15-02757-f002:**
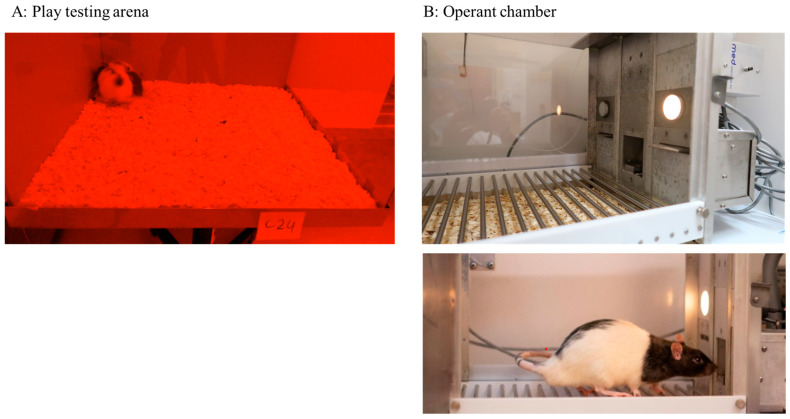
The setups for behavioural assessment. (**A**): Testing arena for social play behaviour. The surface area for play is 40 × 40 cm, covered with 2 cm of wood chips. The sides (two opaque and two clear) have a height of 60 cm to prevent animals from escaping the arena. (**B**): Operant chamber. Retractable levers with cue lights are placed on both sides of the receded pellet dispenser for retrieving sucrose pellets (reward).

**Figure 3 animals-15-02757-f003:**
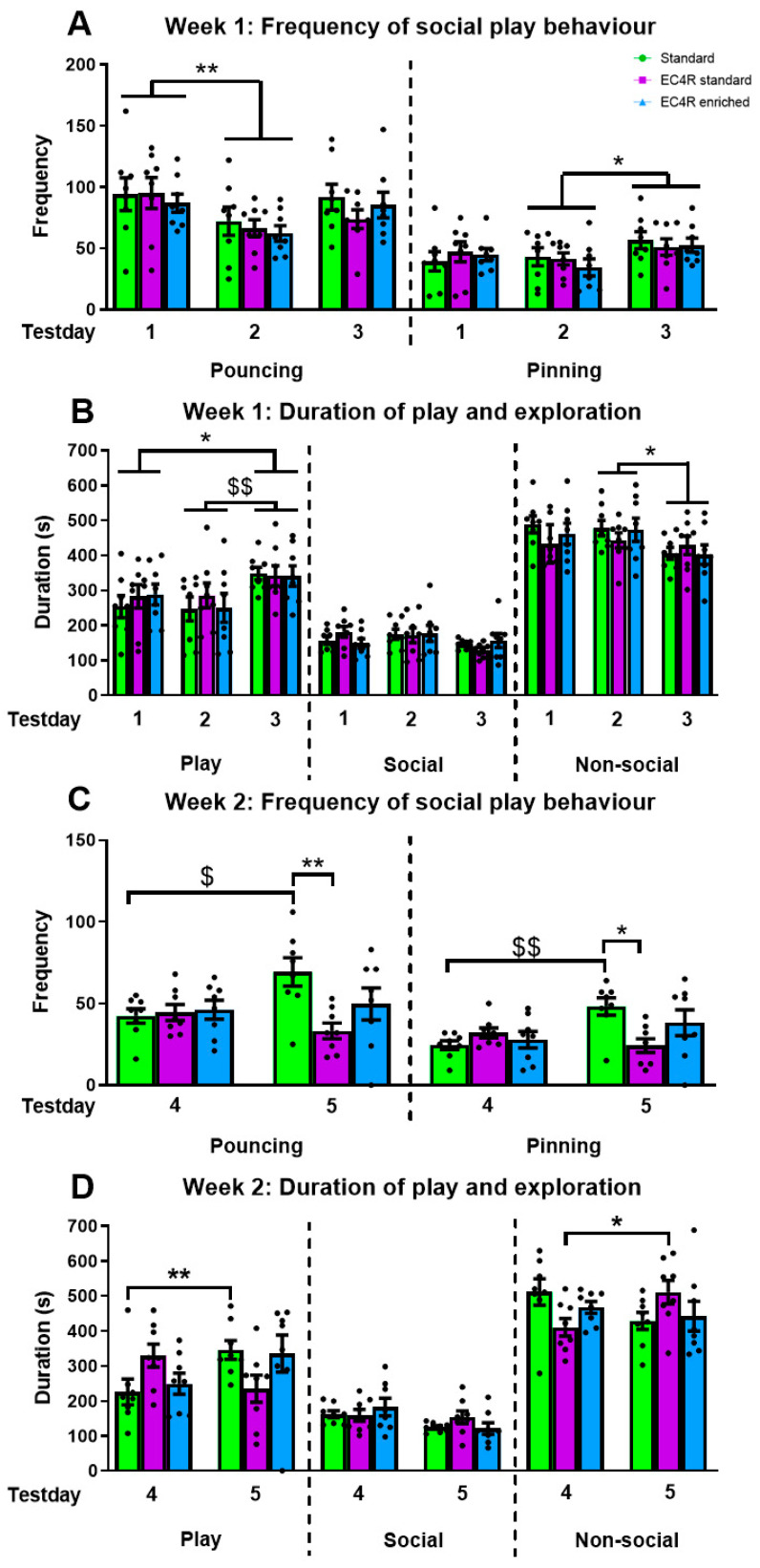
Summarizes the observed play behaviour data for the rats housed in standard cages (green), EC4Rats cages with standard enrichment (purple) or EC4Rats cages with EC4Rats enrichment (blue), respectively. (**A**): Frequency of pounces (play initiation) and pins (continuation play bout) for the three test days (post-natal day 27, 29 and 31) in week 1 of the play peak. Rats from the different housing conditions played similarly across the three test days. Compared to day 1, rats pounced less than day 2 and pinned less on day 2 compared to day 3. (**B**): The time spent engaged in play, social exploration or cage exploration did not differ between housing conditions. Rats spent more time playing on day 3 compared to the other days. Rats spent less time exploring the test cage on day 3 compared to day 2. Social exploration did not differ over days. (**C**): Frequency of pounces and pins in week 2 of the play peak in rats (post-natal day 34 and 36, test day 4 and 5). Rats in standard housed cages pounced and pinned more on day 5 compared to day 4. In addition, on day 5, standard housed rats pounced and pinned more than rats housed in EC4Rats cages with standard enrichment. (**D**): The duration of play behaviour in standard housed rats was higher on day 5 compared to 4, whereas rats in the other housing conditions played similar amounts of time. No differences were found in the duration of social exploration. The non-social exploration time was higher in rats housed in EC4Rats cages with standard enrichment on day 5 compared to 4, whereas rats in the other housing conditions did not show any differences. Data are shown as average ± SEM. Black dots represent the dyads of rats. **/$$ *p* < 0.01 compared to indicated group, */$ *p* < 0.05 compared to indicated group.

**Figure 4 animals-15-02757-f004:**
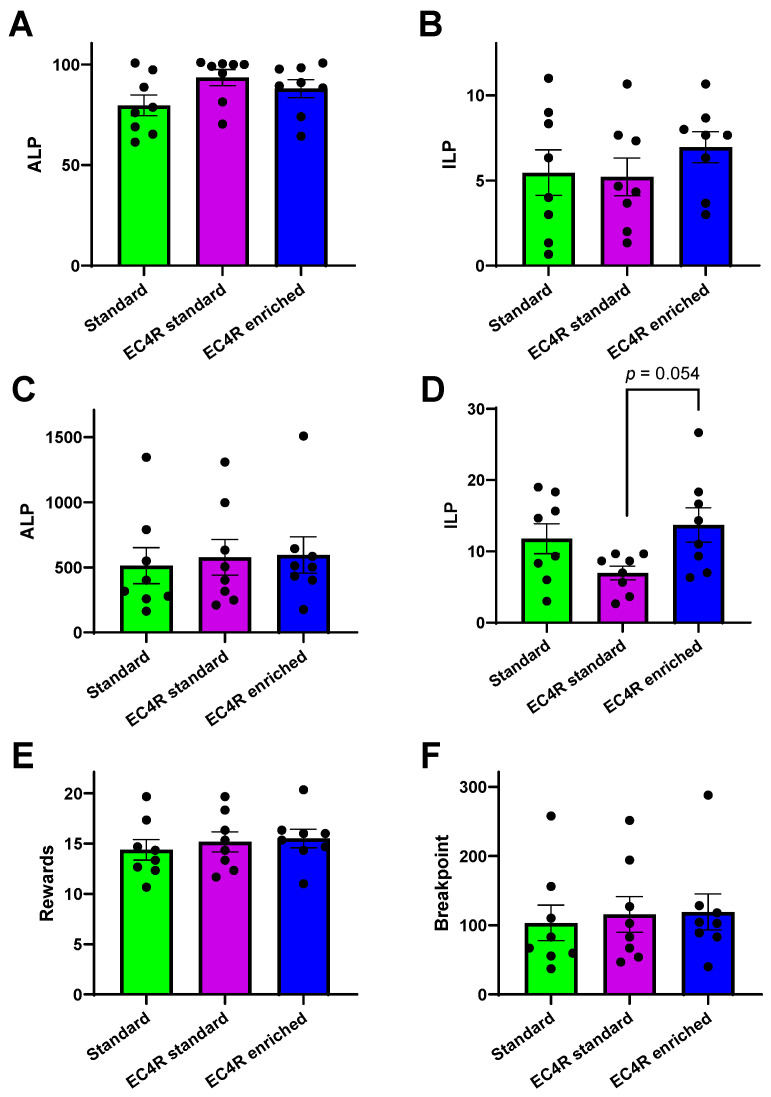
Summarizes the operant sucrose self-administration data for the rats housed in standard cages, EC4Rats cages with standard enrichment or EC4Rats cages with EC4Rats enrichment, respectively. The upper row represents the group averages for (**A**) the number of active lever presses (ALP) and (**B**) the number of inactive lever presses under a fixed ratio 1 schedule of reinforcement. The middle and bottom row summarize the group averages for (**C**) the number of active lever presses (ALP), (**D**) the number of inactive lever presses, (**E**) the number of rewards earned and (**F**) the breakpoint under a progressive ratio schedule of reinforcement. Data are shown as average ± SEM.

## Data Availability

Raw data are available upon request.

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
