# Peer review of "Evaluation of the Effects of Enlarged Housing on Social Play and Reward Seeking in Rats"

_animals, 2025, doi:10.3390/ani15182757_

Round 1
Reviewer 1 Report
Comments and Suggestions for Authors
- I see that two effects are intertwined and cannot be separated: the increase in cage space, which by itself can elicit responses, and the addition of objects (enrichment) inside the cage. Since there was no group that experienced only an increase in cage space without any objects, I suggest that the authors review all interpretations regarding the effect of space, as it was not purely evaluated. For example, reconsider statements like the one in line 369: ‘The current findings suggest that housing rats in larger cages, with or without additional cage enrichment, does not substantially change social play behaviour frequency and duration or sucrose self-administration both under fixed ratio and progressive ratio schedules of reinforcement.’ In fact, the study only includes the condition of housing rats in larger cages with additional cage enrichment.
- The authors omitted citing several studies that also investigate the effects of larger cages. Given the relatively limited number of such studies, it would be valuable to acknowledge other researchers who have contributed to the discussion on rodent housing space and its implications. Some relevant examples include:
- DOI: 10.1258/0023677052886457
- DOI: 10.1139/apnm-2014-0416
- DOI: 10.3389/fnbeh.2019.00221
- DOI: 10.1007/BF00426903
- DOI: 10.1177/00236772211065915
- The behavioral assessments were conducted in an environment different from the one in which the animals live. Considering that large cages with enrichment can also increase the rodents’ spontaneous physical activity in their home cages, do the authors have any additional results or observations to report in this regard?
- In line 196: ‘In case of a significant interaction effect, this was followed by one-way ANOVAs or unpaired Student’s t-tests with Bonferroni correction.’ Please explain this more clearly, because if there is a significant interaction, it is usually sufficient to examine the post hoc results. I do not understand the rationale for performing a one-way ANOVA or t-test in this context.
- Just as you showed photos of the cages, please also include photos of the tests, such as the behavioural assessment of operant conditioning for sucrose and the assessment of social play behaviour.
- In your article, please highlight the precautions taken with the animals, which I believe were implemented and are crucial to avoid disrupting social hierarchy and stressing the animals. For example, animals from the same cage were never exchanged, as social relationships were already established; intruder animals were not introduced into an existing cage; and food was always provided ad libitum to prevent social stress and fights over resources.
- The legend for Figure 2 is too long, and the statistical symbols need to be described in more detail, as they are currently confusing. For example (panel A), you wrote in legend ‘Rats in EC4Rats cages with standard enrichment pounced less than rats in standard housing and pinned less than rats in EC4Rats cages with EC4Rats enrichment’, but from what I understood, all groups on day 2 had lower values in comparison to test day 1.
- Instead of leaving a very lengthy results section as text, consider presenting the data in tables as well. This enriches the results section and improves readability.
- I suggest splitting Figure 2 into separate parts so that each figure has a legend fitting entirely on one page. One figure could present the data for week 1, and another (Figure 3) could present the data for week 2.
- Your statistical analysis appears incorrect. I agree that ANOVA results should report the F and p values, but including the F value for post hoc tests is inappropriate — for post hoc comparisons, reporting the p value (and preferably pared effect sizes) is sufficient.
- You started your interventions with very young, recently weaned animals, continuing approximately until adolescence. If the animals had been kept in the cages for a longer period, you might have observed more adaptations. I think the duration was too short for the cage environment to produce its full effects. Please consider discussing this point.
Reviewer 2 Report
Comments and Suggestions for Authors
The study compared social play behavior and operant response to sucrose in rats housed in larger housing systems to those housed in conventional cage systems. This research is very interesting because there has been a long-standing debate about whether environmental enrichment and non-standard cages can interfere with the reproducibility of experiments.
The data obtained with this research demonstrate that housing rats in larger cages did not affect the reinforcing effects of sucrose nor did it impact their motivation to seek sucrose. It is therefore possible to use enriched cages for this type of experiments, as prescribed by the 3R principle.
Personally, I found the paper well-written, clear, and logical. The introduction is clear and complete. I would add a drawing of the cage prototype to the materials and methods section because it's unclear from the two photographs how it was constructed.
The results are presented clearly, with the aid of graphs, and are discussed appropriately. The conclusions are consistent with the results obtained.
In conclusion, I believe the paper can be accepted for publication in its current form, with the small modification I suggested.
The paper is very interesting for all researchers who use rodents in scientific research.
Reviewer 3 Report
Comments and Suggestions for Authors
General comments
The environmental conditions in which animals live are crucial to maintaining animal welfare, and this is perhaps reflected in environmental enrichment techniques that help promote natural animal behavior. However, there has been little evidence on the living space needed to maintain such behaviors. For this reason, the proposal in this manuscript is appropriate for this journal and innovative. However, one weakness is the lack of a clear objective that allows for a clear link between the title, objective, and conclusions.
Response:
On the other hand, another weakness is the justification for the study, as it is unclear what the state of the art is and what information gap the article aims to fill. Perhaps this is due to the overly broad description in the introduction, which does not adequately position the reader about the problem the authors aim to solve.
Response:
Particular comments
Line 2. I agree with the proposed title of your article; however, if the authors allow me, I suggest that you modify this proposal to “evaluation of the effects of enlarged housing on social play and reward seeking in rats.”
Response:
Line 10. Please mention what type of behavior was evaluated.
Response:
Line 18. Correct the spelling error in the space.
Response:
Lines 20-21. Does this sentence describe your objective? Since it differs from your objective described in lines 65-66, please standardize your objective to avoid confusion for the reader.
Response:
Lines 22-23. The description of your methodology is too vague. If the authors allow me, I suggest that they briefly describe the number of animals, breed, average age, and sex, as well as the general characteristics of their study.
Response:
Lines 24-26. Again, they do not mention what the overall results of their study were, i.e., they evaluated gaming behaviors, so which behaviors increased with the larger space compared to the standard space? Was this increase significant?
Response:
Line 29. If the authors allow me, I suggest that they include the term “animal welfare” in their keywords, as this could increase the likelihood of matches between different databases.
Response:
Line 31. In line with my general comment, I suggest that the authors be more concise in this section, which helps to position the reader on the current issue they intend to address and justifies their study. For example, I suggest that they consider whether it is necessary to address social play behavior independently, as this may be considered repetitive and is not linked to the rest of their topic. As a possible solution, I suggest that this section be shorter and that the information on the need for space be integrated with the development of social play behavior.
Response:
Lines 32-33. I agree with this opening paragraph, but if the authors will allow me, I suggest that the needs of animals to maintain animal welfare be addressed first. In addition, I suggest that they mention the importance of natural behavior expression and whether this is related to a good level of welfare.
Response:
Line 35. Please correct the typographical error.
Response:
Line 36. Please improve the transition in the text, as there is a jump in ideas.
Response:
Line 41. Please add a reference. In addition, if the authors agree, I suggest that after this sentence, they mention the consequences of limited space on behavior.
Response:
Lines 61-64. Why do the authors discuss environmental enrichment? I understand and share the authors' view, but this leap in ideas may confuse the reader if the central theme is the discussion of the importance of living space that allows for the expression of natural behaviors. I suggest mentioning here the gap that your study aims to fill.
Response:
Lines 65-70. Again, there is a lack of connection between the objective and the title, as the title only mentions that they will evaluate the effect of space on play behavior, but here they mention that they will compare behavior. This error may confuse the reader about what you are trying to convey. If the authors allow me, I suggest that you modify this point to “The objective of this study was to evaluate the effects of enlarged housing on social play behavior in rats.” Perhaps this could improve the connection between your title and your objective. In addition, I suggest that you add a hypothesis that complements your objective.
Response:
Lines 71-96. In addition to my general comment, I suggest that the authors explore the need to keep these paragraphs or integrate these ideas into the rest of the text.
Response:
Line 98. Before this section, I suggest adding a subtopic called “ethical statement” where you indicate whether this project was approved by an ethics committee at any institution.
Response:
Line 105. Could you indicate the average age and average weight of the animals?
Response:
Line 107. Was the division of the animals into different types of housing done randomly?
Response:
Lines 348-350. I understand that you mentioned your objective again; however, this sentence may be weak for the beginning of your discussion, so I suggest that you mention the most relevant result of your study.
Response:
Line 354. Could you provide a biological explanation for this result?
Response:
Line 359. Again, I suggest that you provide a biological explanation for this observation.
Response:
Line 365. I understand and find this idea very interesting, but could you elaborate on the fact that it is not enough to increase the space available to promote play behavior; it is also necessary to promote the motivation for this behavior. But could you explain why?
Response:
Lines 378-387. Please add references.
Response:
Round 2
Reviewer 1 Report
Comments and Suggestions for Authors
Thank you for your detailed reply. I fully understand that mandatory enrichment is ethically unavoidable and therefore necessarily part of all housing conditions. However, I believe this raises an important point for the interpretation of the results.
Because enrichment was always present, the effect of increased cage space cannot be regarded as entirely independent. In fact, it is possible that the influence of space was masked by or interacted with the mandatory enrichment provided in all groups. For this reason, the statement that “larger cages, with or without enrichment, do not substantially change behavior” may overstate the conclusions that can be drawn from the design.
A more precise interpretation would be that, under conditions of mandatory enrichment, increasing cage size did not produce detectable changes in social play or sucrose self-administration. This wording would make it clear that the study did not evaluate a condition of “larger cages without enrichment,” and that potential interactions between space and even minimal enrichment cannot be fully excluded.
By acknowledging this nuance, the conclusions will remain fully consistent with the data while also transparently addressing the limitations imposed by ethical requirements.
Author Response
Comment 1:
Thank you for your detailed reply. I fully understand that mandatory enrichment is ethically unavoidable and therefore necessarily part of all housing conditions. However, I believe this raises an important point for the interpretation of the results.
Because enrichment was always present, the effect of increased cage space cannot be regarded as entirely independent. In fact, it is possible that the influence of space was masked by or interacted with the mandatory enrichment provided in all groups. For this reason, the statement that “larger cages, with or without enrichment, do not substantially change behavior” may overstate the conclusions that can be drawn from the design.
A more precise interpretation would be that, under conditions of mandatory enrichment, increasing cage size did not produce detectable changes in social play or sucrose self-administration. This wording would make it clear that the study did not evaluate a condition of “larger cages without enrichment,” and that potential interactions between space and even minimal enrichment cannot be fully excluded.
By acknowledging this nuance, the conclusions will remain fully consistent with the data while also transparently addressing the limitations imposed by ethical requirements.
Response: we have now changed the wording in the abstract, the introduction and the discussion: ..with or without... has been changed to larger cages with mandatory or with additional enrichment.
Reviewer 3 Report
Comments and Suggestions for Authors
General comments
I appreciate the authors for considering the comments suggested above, from a study that I am convinced actively contributes to the field of animal welfare and behavior. However, I still observe minor changes in their proposal that should be corrected before publication.
Response:
Response:
Particular comments
Line 2. Correct the spelling error.
Response:
Line 27. Correct the missing space between words.
Response:
Section 2 Materials and Methods. Please check the numbering format of the subtopics suggested by the author's guide, as they indicate that they should be numbered according to the section they represent.
Response:
Line 241. Could you please specify the assumptions you considered when estimating the sample size? Especially considering the statistical power.
Response:
Reference. Please check the font style used, as it differs from that recommended by the author guidelines.
Response:
Author Response
Line 2. Correct the spelling error.
Response: we do not see a spelling error in line 2?
Line 27. Correct the missing space between words.
Response: done
Section 2 Materials and Methods. Please check the numbering format of the subtopics suggested by the author's guide, as they indicate that they should be numbered according to the section they represent.
Response: done
Line 241. Could you please specify the assumptions you considered when estimating the sample size? Especially considering the statistical power.
Response: we have added a sentence on the assumption of normal distribution
Reference. Please check the font style used, as it differs from that recommended by the author guidelines.
Response: done